# The prenyltransferase UBIAD1 is the target of geranylgeraniol in degradation of HMG CoA reductase

**Marc M Schumacher[1][†], Rania Elsabrouty[1][†], Joachim Seemann[2], Youngah Jo[1], Russell A DeBose-Boyd[3]\***

[1]Department of Molecular Genetics, University of Texas Southwestern Medical Center, Dallas, United States; [2]Department of Cell Biology, University of Texas Southwestern Medical Center, Dallas, United States; [3]Department of Molecular Genetics, Howard Hughes Medical Institute, University of Texas Southwestern Medical Center, Dallas, United States

**Abstract** Schnyder corneal dystrophy (SCD) is an autosomal dominant disorder in humans characterized by abnormal accumulation of cholesterol in the cornea. SCD-associated mutations have been identified in the gene encoding UBIAD1, a prenyltransferase that synthesizes vitamin $K_2$. Here, we show that sterols stimulate binding of UBIAD1 to the cholesterol biosynthetic enzyme HMG CoA reductase, which is subject to sterol-accelerated, endoplasmic reticulum (ER)-associated degradation augmented by the nonsterol isoprenoid geranylgeraniol through an unknown mechanism. Geranylgeraniol inhibits binding of UBIAD1 to reductase, allowing its degradation and promoting transport of UBIAD1 from the ER to the Golgi. CRISPR-CAS9-mediated knockout of UBIAD1 relieves the geranylgeraniol requirement for reductase degradation. SCD-associated mutations in UBIAD1 block its displacement from reductase in the presence of geranylgeraniol, thereby preventing degradation of reductase. The current results identify UBIAD1 as the elusive target of geranylgeraniol in reductase degradation, the inhibition of which may contribute to accumulation of cholesterol in SCD.

\*For correspondence: Russell. Debose-Boyd@utsouthwestern. edu

[†]These authors contributed equally to this work

**Competing interests:** The authors declare that no competing interests exist.

**Reviewing editor**: Ramanujan S Hegde, MRC Laboratory of Molecular Biology, United Kingdom

## Introduction

SCD (Schnyder corneal dystrophy) is a rare autosomal dominant eye disease that is characterized by bilateral opacification of the cornea (*Klintworth, 2009*; *Weiss, 2009*). The clinical manifestation of SCD can be apparent early in life, usually within the first decade. Thereafter, opacification of the cornea progresses slowly and ultimately leads to reduced visual acuity, which is postulated to be caused by light scattering (*Weiss, 2009*). The significance of this visual impairment is highlighted by the frequency in which corneal transplant surgery is utilized in treatment of SCD; approximately 50% of SCD patients ≥50 years of age undergo corneal transplant surgery for restoration of normal vision acuity (*Weiss, 2007*). The frequency of the procedure increases to greater than 70% for SCD patients 70 years and older. Analyses of corneas removed from SCD patients during transplantation surgery revealed a marked accumulation of unesterified cholesterol and lesser accumulation of esterified cholesterol and phospholipids (*McCarthy et al., 1994*; *Gaynor et al., 1996*; *Yamada et al., 1998*), suggesting dysregulation of cholesterol metabolism may underlie pathogenesis of the disease. Systemic dyslipidemia has been reported to be associated with some, but not all cases of SCD (*Thiel et al., 1977*; *Brownstein et al., 1991*; *Crispin, 2002*).

In 2007, two groups independently identified SCD-associated mutations in the gene encoding UBIAD1 (UbiA prenyltransferase domain-containing protein-1) (*Orr et al., 2007*; *Weiss et al., 2007*).

**eLife digest** People with a rare genetic disorder called 'Schnyder corneal dystrophy' gradually lose their vision, because their corneas become increasingly cloudy. This cloudiness is caused by a build-up of excessive amounts of cholesterol, and the disorder itself is caused by mutations in a gene that encodes a protein called UBIAD1. Researchers have previously discovered that the UBIAD1 protein is involved in making vitamin $K_2$, but it is not clear how this protein also helps to control cholesterol levels in the cornea.

An enzyme called HMG CoA reductase makes a molecule that is used to make cholesterol and many other similar sterol molecules. A 'feedback loop' operates in cells to control the amount of the reductase and prevent cholesterol from becoming too high or too low. Sterol molecules, together with another molecule called geranylgeraniol, participate in this feedback loop by promoting the destruction of the reductase enzyme. Here, Schumacher et al. reveal a link between UBIAD1 and the reductase that may explain how UBIAD1 contributes to the production of excess cholesterol in patients with Schnyder corneal dystrophy.

The experiments show that, in the presence of sterol molecules, UBIAD1 can bind to HMG CoA reductase to protect the reductase from being destroyed by other proteins. Geranylgeraniol—which stops the UBIAD1 protein from binding to the enzyme—is required to completely destroy the reductase enzyme. However, when UBIAD1 is missing, the reductase enzyme is destroyed even in the absence of geranylgeraniol.

Furthermore, the experiments show that the genetic mutations linked to Schnyder corneal dystrophy lead to the production of versions of the UBIAD1 protein that bind to the reductase enzyme even when geranylgeraniol molecules are present. This prevents the normal breakdown of the reductase enzyme, which could lead to the build up of cholesterol in the cornea of individuals with the disorder.

Schumacher et al.'s findings show that the UBIAD1 protein helps to control the levels of cholesterol in cells by protecting the HMG CoA reductase enzyme from destruction. These findings may aid the development of new therapies to lower cholesterol levels in cells, which may help patients with Schnyder's corneal dystrophy and other conditions caused by high cholesterol levels.

UBIAD1 (also known as TERE1) was first described as being absent or markedly diminished in bladder and prostate tumors (*McGarvey et al., 2001*, *2003*) and belongs to the UbiA superfamily of prenyltransferases (*Heide, 2009*). These enzymes, which are found in a wide variety of species, contain 8–10 transmembrane helices and catalyze transfer of isoprenyl groups to aromatic acceptors, producing a diverse range of molecules including ubiquinone, chlorophylls, hemes, vitamin E, and vitamin K (*Forsgren et al., 2004*; *Nakagawa et al., 2010*; *Bonitz et al., 2011*). Indeed, UBIAD1 catalyzes transfer of the 20-carbon geranylgeranyl group from geranylgeranyl pyrophosphate to menadione (vitamin $K_3$) derived from plant-derived phylloquinone (vitamin $K_1$), generating MK-4 (menaquinone-4, vitamin $K_2$) (*Figure 1*) (*Nakagawa et al., 2010*; *Hirota et al., 2013*). It has also been proposed that UBIAD1 mediates polyprenylation of 4-hydroxybenzoate to produce 3-polyprenyl-4-hydroxybenzoate (PPHB), an intermediate in the synthesis of CoQ10 (coenzyme Q10 or ubiquinone-10) (*Mugoni et al., 2013*).

To date, 24 UBIAD1 mutations have been identified in ~50 SCD families (*Nickerson et al., 2013*; *Nowinska et al., 2014*). Several of these mutations alter amino acid residues that localize to the active site of the UBIAD1 prenyltransferase domain as determined by molecular modeling using structural models of bacterial and archaeal UbiA prenyltransferases (*Bräuer et al., 2004*, *2008*; *Nickerson et al., 2010*; *Cheng and Li, 2014*; *Huang et al., 2014*). Cells from patients harboring four SCD-associated mutations (N102S, D112N, and G177E/R) exhibited reduced MK-4 biosynthetic activity (*Nickerson et al., 2013*). A possible link between UBIAD1 and cholesterol metabolism was provided by co-immunoprecipitation studies that showed an association of UBIAD1 with the cholesterol biosynthetic enzyme HMG CoA (3-hydroxy-3-methylglutaryl coenzyme A) reductase (*Nickerson et al., 2013*). However, the relevance of this association to regulation of cholesterol metabolism and pathogenesis of SCD is not clear.

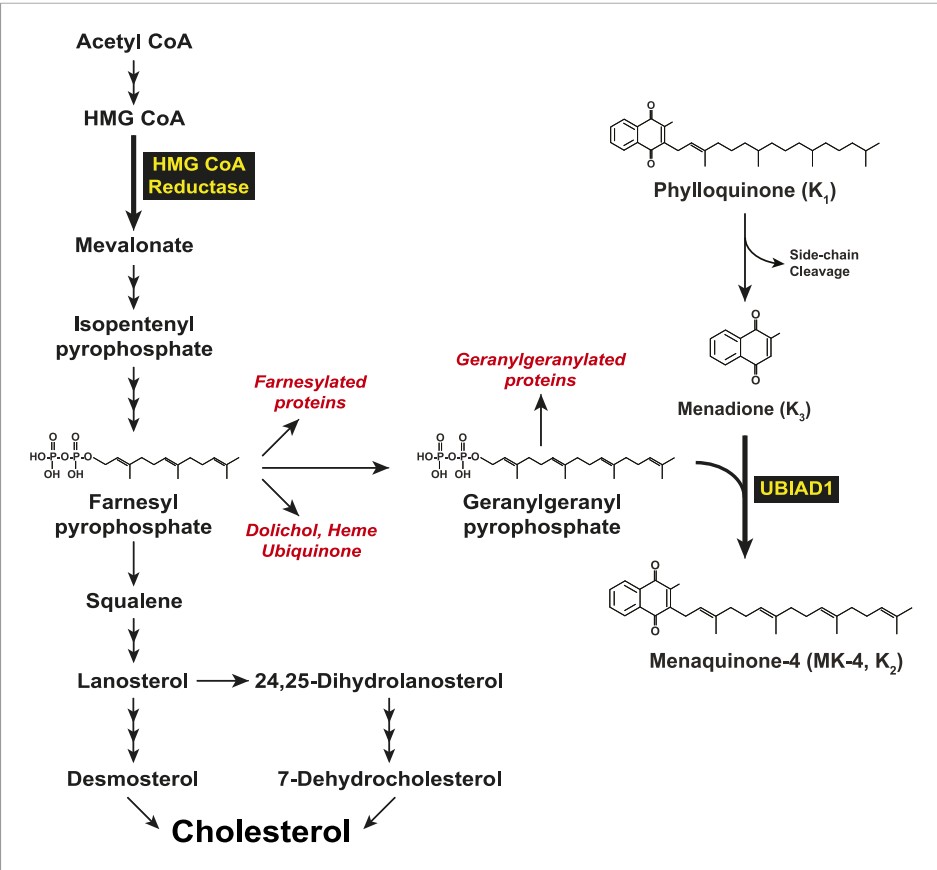

**Figure 1**. Biosynthesis of cholesterol and menaquinone-4 (MK-4, vitamin K₂) in mammalian cells.

In mammalian cells, the ER (endoplasmic reticulum)-localized reductase catalyzes reduction of HMG CoA to mevalonate (*Figure 1*). This reaction constitutes a rate-limiting step in synthesis of cholesterol as well as nonsterol isoprenoids such as the farnesyl and geranylgeranyl groups that are transferred to many cellular proteins and utilized in synthesis of ubiquinone, heme, and dolichol (*Goldstein and Brown, 1990*; *Casey and Seabra, 1996*). Multiple feedback mechanisms converge on reductase to ensure a constant supply of nonsterol isoprenoids, while avoiding overaccumulation of cholesterol (*Brown and Goldstein, 1980*). This feedback system is exploited therapeutically by a group of competitive inhibitors of reductase called statins, which trigger regulatory responses that result in the lowering of plasma levels of LDL (low-density lipoprotein)-cholesterol and thereby reduce the incidence of cardiovascular disease (*Stossel, 2008*; *Marais et al., 2014*). Statins also block production of sterol and nonsterol isoprenoids that govern feedback regulation of reductase (*Goldstein and Brown, 1990*). Thus, a marked accumulation of reductase occurs in livers of statin-treated animals and humans (*Brown and Goldstein, 1980*; *Kita et al., 1980*; *Reihner et al., 1990*), which can blunt the cholesterol-lowering effects of these drugs. This accumulation results in part, from slowed ERAD (ER-associated degradation) of reductase (*Nakanishi et al., 1988*; *Inoue et al., 1991*; *Ravid et al., 2000*).

The ERAD of reductase is initiated by intracellular accumulation of sterols, which causes the enzyme to bind to ER membrane proteins called Insig-1 and Insig-2 (*Sever et al., 2003a*, *2003b*) (*Figure 2*). Insig binding occurs through the membrane domain of reductase, which contains eight membrane-spanning helices and precedes a large cytosolic domain with enzymatic activity (*Liscum et al., 1985*; *Roitelman et al., 1992*). Insig-associated ubiquitin ligases gp78 and Trc8 facilitate ubiquitination of cytosolically exposed lysine residues in the membrane domain of reductase (*Sever et al., 2003a*; *Song et al., 2005*; *Jo et al., 2011*; *Liu et al., 2012*). This ubiquitination marks

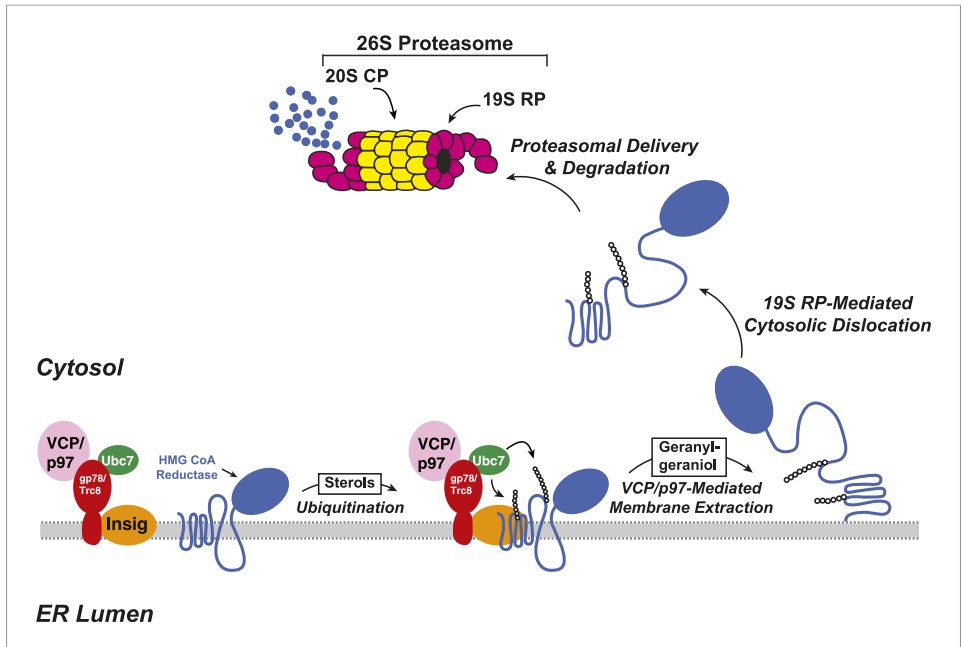

**Figure 2**. Insig-mediated, sterol-accelerated degradation of HMG CoA reductase in mammalian cells.

reductase for recognition by the AAA (ATPases associated with diverse cellular activities)-ATPase VCP/p97, which mediates extraction of reductase across ER membranes (*Morris et al., 2014*). Once extracted, ubiquitinated reductase is then released into the cytosol and delivered into the proteolytic chamber of the 20S proteasome through reactions mediated by the proteasome 19S regulatory particle, which contains six AAA-ATPases (*Ehlinger and Walters, 2013*). Geranylgeraniol, the alcohol derivative of geranylgeranyl pyrophosphate, augments sterol-accelerated ERAD of reductase but does not appreciably affect sterol-induced ubiquitination (*Sever et al., 2003a*). This observation led to the notion that geranylgeraniol augments post-ubiquitination steps in reductase ERAD (*Figure 2*). Indeed, geranylgeraniol enhances sterol-induced membrane extraction and cytosolic dislocation of reductase as judged by assays carried out in vitro and in intact cells (*Song and DeBose-Boyd, 2004*; *Elsabrouty et al., 2013*; *Morris et al., 2014*).

In the current studies, we use proximity-dependent biotinylation (*Roux et al., 2012*) to identify UBIAD1 as an associated protein of the membrane domain of reductase, which is necessary and sufficient for sterol-accelerated ERAD (*Gil et al., 1985*; *Skalnik et al., 1988*). Endogenous UBIAD1 binds to reductase in sterol-treated cells as determined by co-immunoprecipitation. This sterol-induced binding is inhibited by geranylgeraniol, releasing UBIAD1 for translocation from membranes of the ER to the Golgi. CRISPR/Cas9-mediated knockout and RNA interference (RNAi)-mediated knockdown of UBIAD1 relieves the geranylgeraniol requirement for reductase ERAD. UBIAD1 harboring the SCD-associated N102S mutation resists geranylgeraniol-mediated displacement from reductase and remains associated with the protein in the ER, thereby blocking sterol-accelerated degradation of reductase. These studies identify UBIAD1 as the elusive target of geranylgeraniol in the ERAD of reductase and provide insight into mechanisms through which the nonsterol isoprenoid modulates the reaction. Moreover, our results suggest that slowed ERAD of reductase may contribute to the accumulation of cholesterol observed in SCD.

## Results

To identify proteins that associate with reductase, we employed a novel chimeric protein composed of the membrane domain of reductase (amino acids 1–346) fused to the promiscuous biotin ligase BirA*, which features proximity-dependent biotinylation of the chimera's near-neighbor proteins (*Roux et al., 2012*). Cells stably expressing high levels of the reductase-BirA* chimera were subjected to treatment with the oxysterol 25-HC (25-hydroxycholesterol) in the presence of biotin.

Following treatments, cells were harvested and lysed; the lysates were then incubated with streptavidin-conjugated agarose beads to capture biotinylated proteins that were subsequently identified by tandem mass spectrometry.

*Figure 3—figure supplement 1B* shows an experiment in which we used co-immunoprecipitation to measure the interaction of endogenous reductase with several proteins that were identified as associated with the reductase-BirA* chimera (*Figure 3—figure supplement 1A*). SV-589 cells, a line of transformed human fibroblasts (*Yamamoto et al., 1984*), were depleted of sterols and subsequently treated in the absence or presence of 25-HC. Following treatments, the cells were harvested for lysis in detergent-containing buffer; the resulting lysates were then immunoprecipitated with polyclonal antibodies against reductase. Immunoblot analysis revealed that treatment of cells with 25-HC triggered co-immunoprecipitation of endogenous reductase with Insig-1 (*Figure 3—figure supplement 1B*, lanes 3 and 4); the control ER membrane protein calnexin failed to interact with reductase, regardless of the absence or presence of 25-HC (lanes 5 and 6). ERGIC-53 similarly failed to appear in the anti-reductase immunoprecipitates (lanes 9 and 10). In contrast, lamina-associated peptide-2 (lanes 7 and 8), peroxiredoxin-4 (lanes 11 and 12), PGRMC2 (lanes 13 and 14), and annexin-A1 (lanes 15 and 16) co-precipitated with reductase in both the absence and presence of 25-HC; the significance and specificity of these interactions are unclear. In the absence of 25-HC, a small amount of UBIAD1 (UbiA prenyltransferase domain-containing protein-1) appeared in the pellet fraction of the reductase immunoprecipitation (lane 17); this appearance was markedly enhanced when the cells were subjected to treatment with 25-HC (lane 18).

UBIAD1 belongs to the UbiA superfamily of prenyltransferases (*Heide, 2009*; *Bonitz et al., 2011*). Similar to other UbiA prenyltransferases, UBIAD1 is very hydrophobic; the human enzyme is comprised of 338 amino acids and is predicted to contain at least eight transmembrane helices (*Figure 3A*). Besides its regulated association with reductase (this study and *Nickerson et al., 2013*), we chose UBIAD1 for further examination owing to the observation that mutations in the UBIAD1 gene are associated with SCD, which may result from dysregulated cholesterol metabolism (*McCarthy et al., 1994*; *Gaynor et al., 1996*; *Yamada et al., 1998*). The experiment of *Figure 3B* shows a time course of 25-HC-mediated association of UBIAD1 with reductase as determined by co-immunoprecipitation. Immunoblot analysis of anti-reductase immunoprecipitates revealed co-precipitation of endogenous UBIAD1 with reductase following treatment of cells for 30 min with 25-HC (*Figure 3B*, top panel, lane 6). This sterol-induced co-precipitation continued after prolonged incubation (top panel, lanes 7–14), even though the total amount of reductase was reduced by accelerated degradation (second and fourth panels, lanes 12 and 14). The structurally related 1,1-bisphosphonate esters SR-12813 and Apomine mimic 25-HC in accelerating reductase ubiquitination and degradation (*Roitelman et al., 2004*; *Sever et al., 2004*; *Nguyen et al., 2009*). As shown in *Figure 3C*, Apomine triggered co-precipitation between reductase and UBIAD1 in a manner similar to that of 25-HC (second panel, compare lane 2 with lanes 3–5).

We next used RNA interference (RNAi) to examine a role for Insigs in the 25-HC-induced binding of reductase to UBIAD1. SV-589 cells were transfected with duplexes of small interfering RNAs (siRNAs) against mRNAs encoding green fluorescent protein (GFP), which is not expressed in the cells, or Insig-1 and Insig-2. Following transfection, the cells were depleted of sterols, incubated in the absence or presence of 25-HC, and lysed for subsequent immunoprecipitation with polyclonal anti-reductase. Immunoblotting of precipitated proteins revealed that 25-HC stimulated co-precipitation of both UBIAD1 and Insig-1 with reductase in SV-589 cells transfected with control GFP siRNA (*Figure 4A*, second and third panels, lane 2). The sterol-induced co-precipitation of UBIAD1 with reductase was markedly reduced in cells that received siRNAs against Insig-1 and Insig-2 (second panel, lanes 3 and 4). The experiment of *Figure 4B* shows that knockdown of UBIAD1 reduced, but did not eliminate the sterol-induced binding of reductase to Insig-1 (third panel, compare lanes 2 and 4). Finally, we examined the requirement of reductase for the association of UBIAD1 with Insig-1. Immunoblot analysis of anti-UBIAD1 immunoprecipitates revealed that as expected, reductase and Insig-1 co-precipitated with UBIAD1 when control siRNA-transfected cells were treated with 25-HC (*Figure 4C*, top and bottom panels, lane 2). However, RNAi-mediated knockdown of reductase abolished co-precipitation of Insig-1 with UBIAD1 (bottom panel, compare lane 4 with lane 2).

Results of *Figure 3B* show that UBIAD1 continued to appear in anti-reductase immunoprecipitates after prolonged treatment with 25-HC, even though total reductase was reduced by accelerated degradation. We reasoned that the proteasome inhibitor MG-132 would block this degradation,

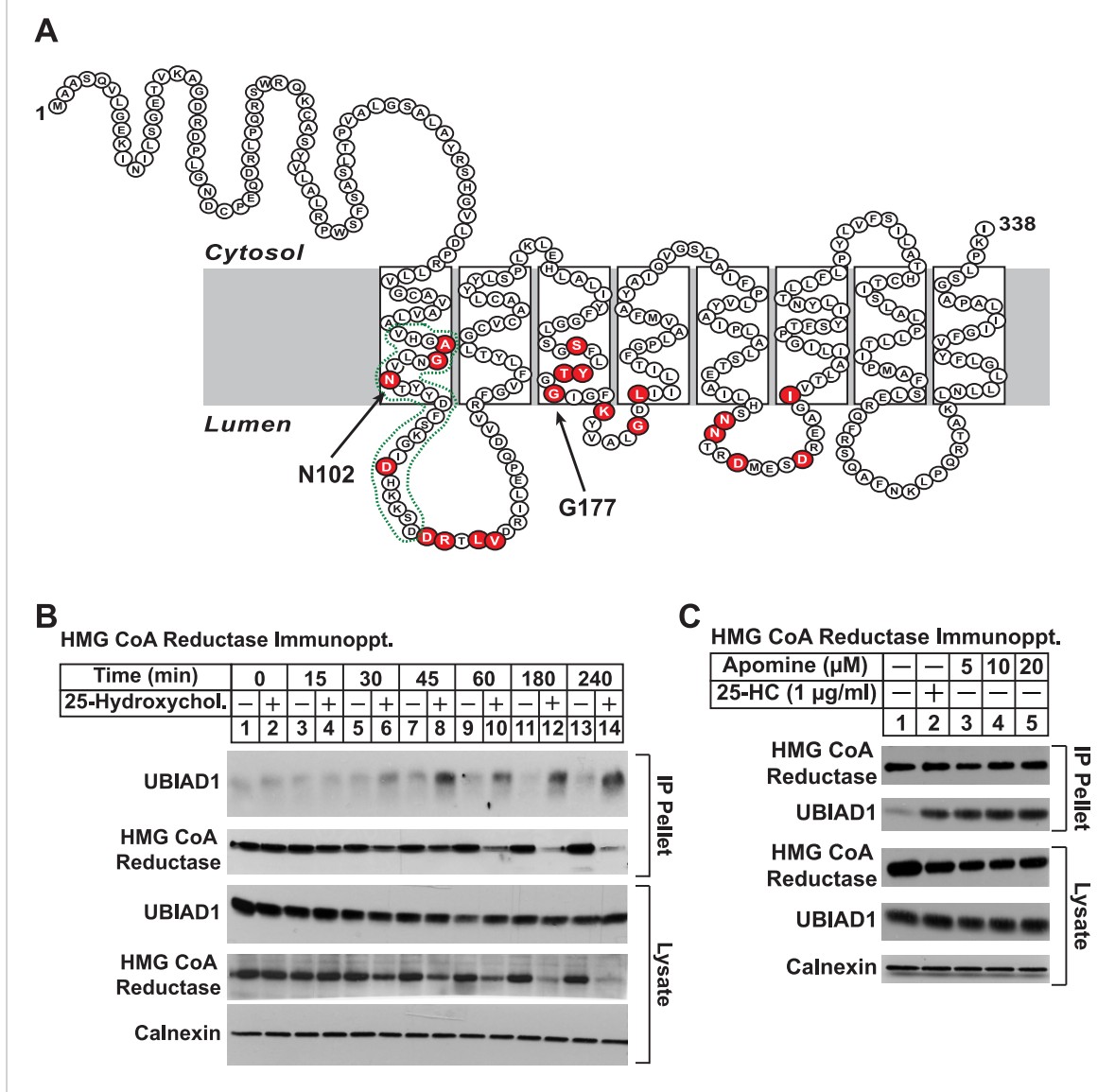

**Figure 3**. Identification of UBIAD1 as an associated protein of HMG CoA reductase. (**A**) Amino acid sequence and predicted topology of human UBIAD1. Amino acid residues mutated in Schnyder corneal dystrophy (SCD) families are shaded in red; asparagine-102 (N102) and glycine-177 (G177), the two most frequently altered UBIAD1 residues in SCD, are indicated by arrows. The predicted active site in the prenyltransferase domain of UBIAD1 is indicated by the green dotted line. (**B** and **C**) SV-589 cells were set up for experiments on day 0 at a density of $2 \times 10^5$ cells per 100-mm dish in medium A supplemented with 10% FCS. On day 3, cells were switched to medium A containing 10% NC-LPPS, 10 µM sodium compactin, and 50 µM sodium mevalonate to deplete sterols. (**B**) After 16 hr at 37°C, the cells received sterol-depleting medium in the absence or presence of 1 µg/ml 25-HC and were further incubated for the indicated period of time 37°C. Cells were then harvested, lysed in PBS containing 1% digitonin, and the resulting lysates were subjected to immunoprecipitation with polyclonal anti-reductase antibodies as described in 'Materials and methods'. Aliquots of precipitated material (IP Pellet) and lysates were subjected to SDS-PAGE and immunoblot analysis was carried out with IgG-H8 (against UBIAD1), IgG-A9 (against reductase), IgG-17H1 (against Insig-1), and anti-calnexin IgG. (**C**) Following sterol depletion, cells were incubated for 45 min at 37°C in sterol-depleting medium containing the indicated concentration of Apomine. Cells were then harvested, lysed, and immunoprecipitated with polyclonal anti-reductase; aliquots of precipitated material and lysates were analyzed by SDS-PAGE, followed by immunoblot as described in (**B**).

The following figure supplement is available for figure 3:

**Figure supplement 1**. Identification of proteins associated with HMG-Red(TM1-8)-BirA*.

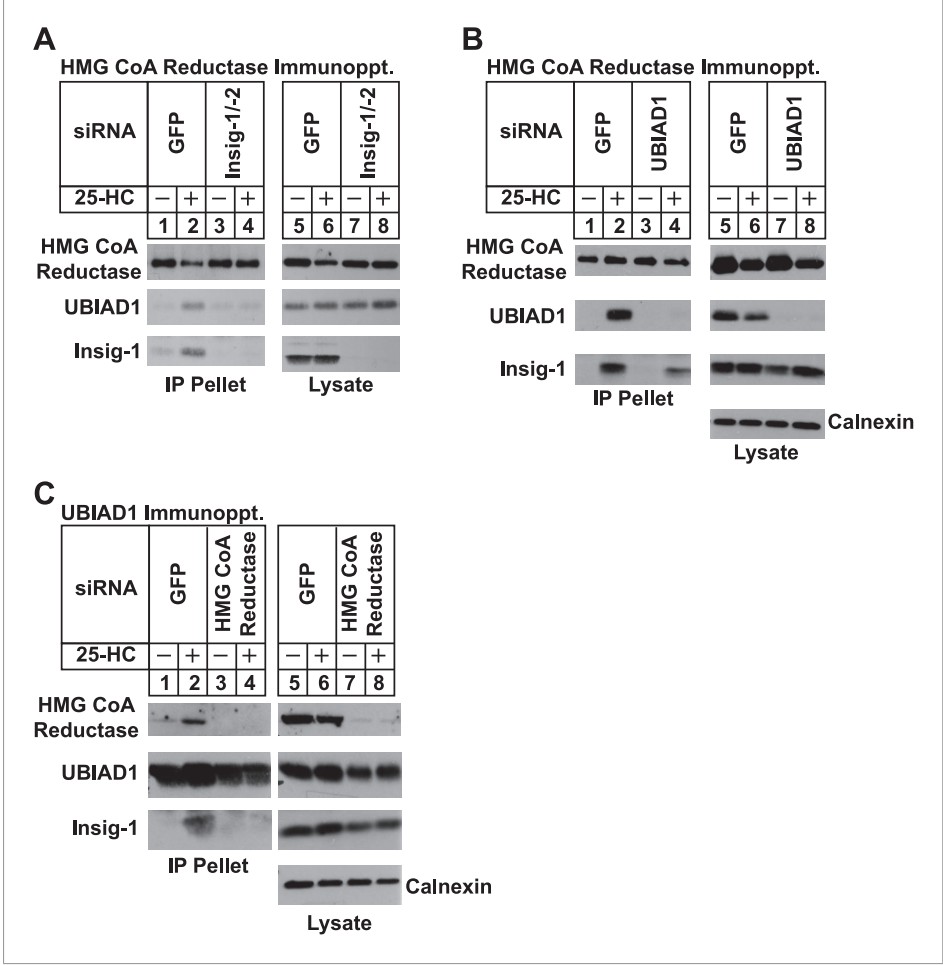

**Figure 4.** Specificity of sterol-dependent UBIAD1-HMG CoA reductase association. SV-589 cells were set up on day 0 at 1 × 10⁵ cells per 100-mm dish in medium A containing 10% FCS. On day 3, cells were transfected with siRNAs targeting mRNAs encoding GFP, Insig-1 and Insig-2, UBIAD1, or reductase as indicated and described in 'Materials and methods'. Cells transfected with siRNA duplexes against reductase received 200 mM mevalonate to provide essential nonsterol isoprenoids. On day 4, cells were depleted of sterols through incubation for 16 hr at 37°C in medium A containing 10% NC-LPPS, 10 μM compactin, and 50 μM mevalonate. The cells then received identical medium in the absence or presence of 1 μg/ml 25-HC. After 45 min at 37°C, cells were harvested, lysed, and immunoprecipitated with polyclonal antibodies against either reductase (**A** and **B**) or UBIAD1 (**C**). The resulting precipitated material and lysates were subjected to SDS-PAGE and immunoblot analysis with IgG-A9 (against reductase), IgG-H8 (against UBIAD1), IgG-17H1 (against Insig-1), and anti-calnexin IgG.

leading to increased co-immunoprecipitation of UBIAD1 with stabilized reductase. Immunoblot analysis of anti-reductase immunoprecipitates from lysates of cells treated in the absence of MG-132 shows that 25-HC stimulated reductase degradation (*Figure 5—figure supplement 1A*, top panel, compare lanes 1 and 2), which was blocked by MG-132 (lanes 3 and 4). Insig-1 co-precipitated with reductase in the presence of 25-HC (second panel, lane 2) and this co-precipitation was enhanced by MG-132 (lane 4). UBIAD1 also co-precipitated with reductase in 25-HC-treated cells (third panel, lane 2); however, the interaction was reduced, rather than enhanced, in the presence of MG-132 (lane 4).

Taking into consideration results of *Figure 5—figure supplement 1A* together with the previous observation that geranylgeraniol augments post-ubiquitination steps in reductase ERAD (*Sever et al., 2003a*; *Elsabrouty et al., 2013*), we next sought to determine whether geranylgeraniol modulates UBIAD1-reductase binding. As expected, 25-HC dose-dependently stimulated co-precipitation of UBIAD1 with reductase (*Figure 5A*, top panel, lanes a–c); however, this sterol-regulated association was inhibited in the presence of 10 or 20 μM geranylgeraniol (lanes d–i). Geranylgeraniol-mediated

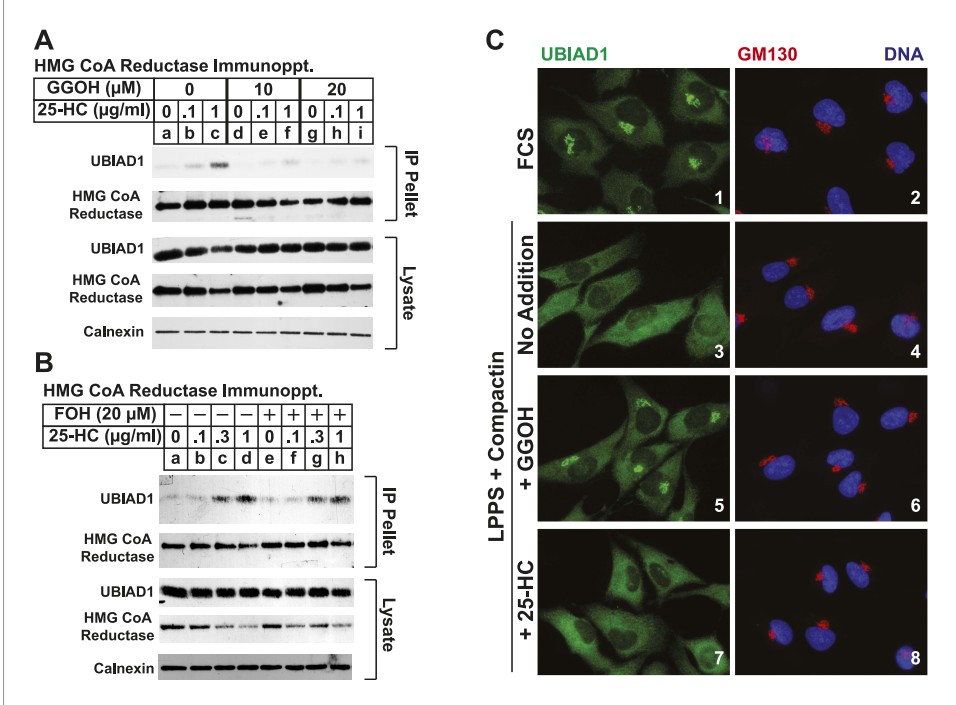

**Figure 5**. The nonsterol isoprenoid geranylgeraniol inhibits sterol-induced binding of UBIAD1 to HMG CoA reductase and promotes its translocation to the Golgi. (**A** and **B**) SV-589 cells were set up for experiments on day 0 and depleted of sterols on day 3 as described in the legend to *Figure 3*. Following sterol-depletion, cells received medium A containing 10% NC-LPPS, 10 μM compactin, 50 μM mevalonate with the indicated concentration of 25-HC in the absence or presence of 10 or 20 μM geranylgeraniol (**A**) or 20 μM farnesol (**B**). Following incubation for 45 min at 37°C, cells were harvested, lysed, and immunoprecipitated with polyclonal antibodies against reductase. Aliquots of the resulting immunoprecipitates and lysates were subjected to immunoblot analysis with IgG-A9 (against reductase), IgG-H8 (against UBIAD1), IgG-17H1 (against Insig-1), and anti-calnexin IgG. (**C**) SV-589 cells were set up on day 0 at 7.5 × 10⁴ cells/well of six-well plates with glass coverslips in medium A containing 10% FCS. On day 1, the cells were switched to identical medium or medium A containing 10% NC-LPPS, 10 μM compactin, and 50 μM mevalonate as indicated. Following incubation for 16 hr at 37°C, the cells were treated in the absence or presence of 30 μM geranylgeraniol (GGOH) or 1 μg/ml 25-HC for an additional 4 hr at 37°C. The cells were subsequently fixed for microscopy as described in 'Materials and methods'.

The following figure supplements are available for figure 5:

**Figure supplement 1**. The proteasome inhibitor MG-132 and geranylgeraniol inhibit sterol-induced binding of UBIAD1 to HMG CoA reductase.

**Figure supplement 2**. Geranylgeraniol, but not 25-HC, FOH, or cholesterol, stimulates translocation of endogenous UBIAD1 to the Golgi in cells deprived of sterol and nonsterol isoprenoids.

**Figure supplement 3**. Geranylgeraniol, but not 25-HC or farnesol, stimulates translocation of transfected UBIAD1 to the Golgi in cells deprived of sterol and nonsterol isoprenoids.

inhibition of reductase-UBIAD1 binding was also observed in the repeat experiment shown in *Figure 5—figure supplement 1B*. In contrast, sterol-induced binding of UBIAD1 to reductase continued in the presence of 20 μM farnesol (*Figure 5B*, top panel, compare lanes a–d with lanes e–h), a 15-carbon nonsterol isoprenoid that does not augment sterol-accelerated ERAD of reductase (*Sever et al., 2003a*).

Consistent with a role in the ERAD of reductase, UBIAD1 has been localized to membranes of the ER (*Nakagawa et al., 2010*; *Nickerson et al., 2013*). However, it should be noted that the prenyltransferase has also been localized to membranes of the mitochondria (*Nickerson et al., 2010*)

as well as the Golgi (*Mugoni et al., 2013*; *Wang et al., 2013*). In the experiment shown in *Figure 5C*, we examined the subcellular localization of endogenous UBIAD1 in SV-589 cells grown in medium containing FCS (fetal calf serum). The results show that endogenous UBIAD1 localized to juxtanuclear structures that resembled the Golgi and also stained with antibodies against the Golgi protein GM130 (*Nakamura et al., 1995*) (*Figure 5C*, panels 1 and 2). The Golgi localization of UBIAD1 was markedly diminished when the cells were depleted of sterol and nonsterol isoprenoids through incubation in medium containing LPPS (lipoprotein-poor serum) and the reductase inhibitor compactin (*Figure 5C*, panels 3 and 4). Notably, localization of Golgi-localized GM130 was unaffected by depletion of sterol and nonsterol isoprenoids (compare panels 2 and 4). The Golgi localization of UBIAD1 was restored in sterol- and nonsterol isoprenoid-depleted cells by the addition of geranylgeraniol (*Figure 5C*, panel 5), but not by the addition of 25-HC (panel 7). A repeat experiment confirms that sterol and nonsterol isoprenoid depletion disrupted the Golgi localization of UBIAD1 (*Figure 5—figure supplement 2*). Importantly, this localization was restored by geranylgeraniol, but not by farnesol, 25-HC, or cholesterol. We obtained similar results with SV-589 cells stably transfected with pCMV-Myc-UBIAD1 encoding Myc-tagged human UBIAD1. Depletion of sterol and nonsterol isoprenoids disrupted Golgi localization of transfected Myc-UBIAD1 (*Figure 5—figure supplement 3*). This localization was restored by geranylgeraniol, but not by farnesol or 25-HC. Notably, endogenous UBIAD1 exhibited relatively more non-Golgi staining compared to Myc-UBIAD1 in FCS-cultured cells, suggesting anti-Myc may have a greater degree of specificity compared to that of anti-UBIAD1.

The role of UBIAD1 in sterol-accelerated reductase ERAD was next examined using RNAi-mediated knockdown and CRISPR/Cas9-mediated knockout (*Cong et al., 2013*; *Mali et al., 2013*). For RNAi experiments, SV-589 cells were transfected with siRNA duplexes targeting the GFP or UBIAD1 mRNAs and subsequently treated in the absence or presence of 25-HC and geranylgeraniol prior to subcellular fractionation. *Figure 6A* shows that 25-HC stimulated reductase degradation from membranes of control siRNA-transfected cells (top panel, lane 2); this degradation was enhanced by geranylgeraniol (lanes 3 and 4). In UBIAD1 knockdown cells, 25-HC stimulated complete degradation of reductase, even in the absence of geranylgeraniol (lanes 5–8). Nearly identical results were obtained using an siRNA duplex targeting a different region of the UBIAD1 mRNA (*Figure 6—figure supplement 1*). Geranylgeraniol also augmented the ERAD of reductase that was stimulated by Apomine (*Figure 6B*, top panel, lanes 1–4); however, the 1,1-bisphosphonate ester caused reductase to become completely degraded in UBIAD1 knockdown cells (lanes 5–8). Results consistent with those observed using RNAi were obtained in cells subjected to CRISPR/Cas9-mediated knockout of UBIAD1. In wild type SV-589 cells, Apomine stimulated reductase ERAD through a mechanism augmented by geranylgeraniol (*Figure 6C*, top panel, lanes 1–4), whereas the compound caused complete degradation of reductase in cells deficient in UBIAD1 (designated UBIAD1⁻) (lanes 5–8). *Figure 6D* shows that in a time-dependent fashion, geranylgeraniol augmented Apomine-induced degradation of reductase in SV-589 cells (compare lanes 1–6 with lanes 7–12). However, Apomine alone caused complete degradation of reductase in UBIAD1⁻ cells (*Figure 6D*, lanes 13–18) and augmentation by geranylgeraniol was markedly diminished (lanes 19–24). Importantly, transfection of UBIAD1⁻ cells with pCMV-Myc-UBIAD1 restored the requirement of geranylgeraniol for maximal degradation of reductase (*Figure 6E*, top panel, compare lanes 1–3 with lanes 4–6).

To date, 24 mutations in the UBIAD1 gene have been identified in SCD families (*Nickerson et al., 2013*; *Nowinska et al., 2014*); these mutations result in the alteration of 20 amino acid residues in UBAID1 (*Figure 3A*). Asparagine-102 (N102), which localizes to the prenyltransferase domain of UBIAD1 (*Cheng and Li, 2014*; *Huang et al., 2014*), is the most frequently altered residue (~30%) in SCD families. Considering that SCD is an autosomal dominant disorder, we reasoned that overexpression of UBIAD1 harboring the SCD-associated N102S mutation would block reductase ERAD. *Figure 7A* shows that when overexpressed in SV-589 cells, the T7-tagged membrane domain of reductase became degraded in the presence of 25-HC and geranylgeraniol (top panel, lanes 1 and 2). Co-expression of a plasmid encoding wild type UBIAD1 stabilized reductase in a dose-dependent fashion and 25-HC plus geranylgeraniol continued to accelerate its ERAD (lanes 3–10). Notably, the highest level of wild type UBIAD1 co-expressed with reductase blunted the enzyme's accelerated degradation (lanes 11 and 12). Co-expression of UBIAD1 (N102S) also stabilized the membrane domain of reductase (*Figure 7B*, top panel, compare lane 1 with lanes 3, 5, 7, 9, and 11), suggesting the protein continued to bind to the enzyme. However, UBIAD1 (N102S) blunted sterol-accelerated ERAD of reductase, even at low levels of expression (*Figure 7B*, top panel, compare lane 2 with

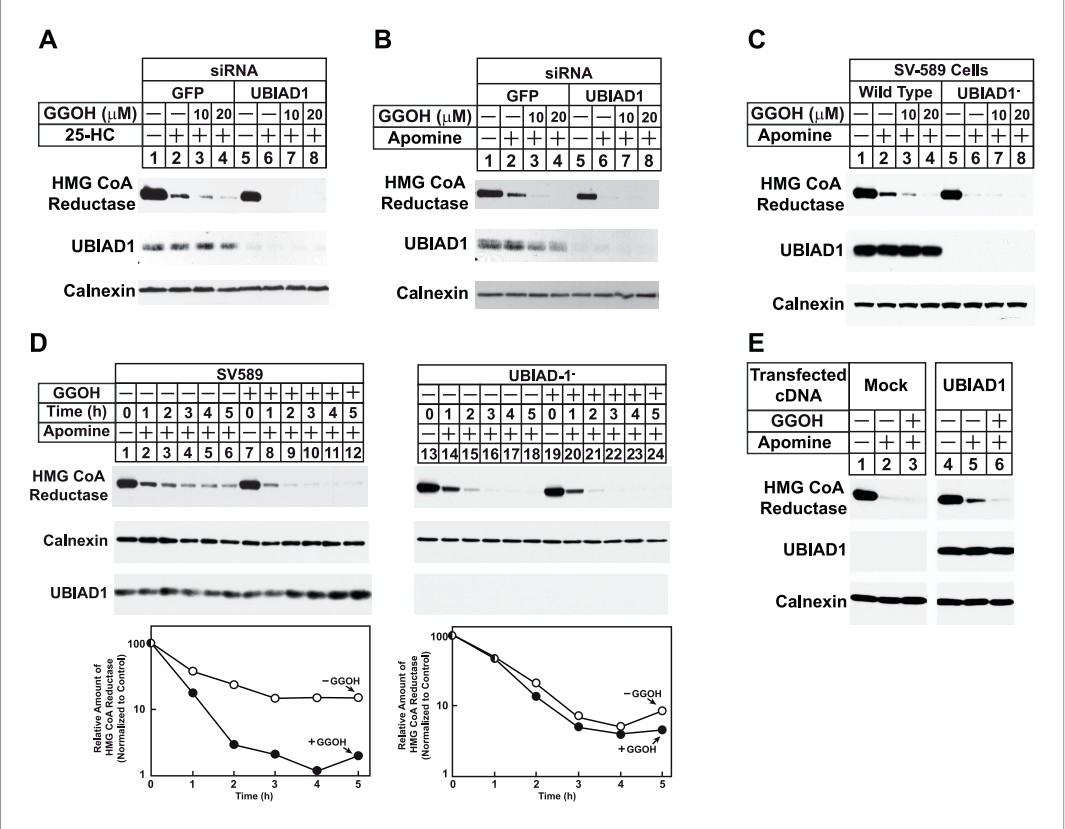

**Figure 6**. RNA interference-mediated knockdown or CRISPR/Cas9-mediated knockout of UBIAD1 alleviates geranylgeraniol requirement in sterol-accelerated degradation of HMG CoA reductase. (**A** and **B**) SV-589 cells were set up for experiments on day 0, transfected with the indicated siRNA duplexes on day 3, and depleted of sterols as described in the legend to *Figure 4*. The sterol-depleted cells were then treated with medium A containing 10% NC-LPPS, 10 μM compactin, and 50 μM mevalonate in the absence or presence of 1 μg/ml 25-HC and geranylgeraniol (GGOH) as indicated. Following incubation for 4 hr at 37°C, cells were harvested for subcellular fractionation. Aliquots of resulting membrane fractions (20 μg protein/lane) were subjected to SDS-PAGE and immunoblot analysis was carried out with IgG-A9 (against reductase), IgG-H8 (against UBIAD1), and anti-calnexin IgG. (**C–E**) SV-589, UBIAD1⁻, UBIAD1⁻/pCDNA3.1, and UBIAD1⁻/pMyc-UBIAD1 cells were set up for experiments on day 0 at $3.5 \times 10^5$ cells per 60-mm dish in medium A containing 10% FCS. On day 1, cells were switched to medium A supplemented with 10% NC-LPPS, 10 μM compactin, and 50 μM mevalonate. Following incubation for 16 hr at 37°C, cells received the identical medium in the absence or presence of 10 μM Apomine and the indicated concentration (**C**) or 10 μM (**D** and **E**) of geranylgeraniol (GGOH). After the indicated period of time (**D**) or 5 hr (**C** and **E**) at 37°C, cells were subjected to subcellular fractionation and membrane fractions (12 μg protein/lane) were analyzed by immunoblot as described in (**A**).
The following figure supplement is available for figure 6:

**Figure supplement 1**. RNA interference-mediated knockdown of UBAD1 alleviates requirement for geranylgeraniol in sterol-accelerated reductase degradation.

lanes 4, 8, 10, and 12). Similar results were obtained with T7-tagged, full-length reductase (*Figure 7—figure supplement 1*). The protein was subjected to sterol-accelerated degradation when transfected alone or together with up to 100 ng of plasmid encoding wild type UBIAD1 (*Figure 7—figure supplement 1A*, top panel, lanes 1–8); co-transfection of higher levels (300 and 1000 ng) of the UBIAD1-encoding plasmid inhibited reductase degradation (lanes 9–12). Consistent with results obtained with the reductase membrane domain, inhibition of full-length reductase degradation was observed upon co-transfection with a significantly lower amount (30 ng) of plasmid encoding UBIAD1 (N102S) (*Figure 7—figure supplement 1B*, top panel, compare lanes 1–4 with lanes 5–12). Glycine-177 (G177, see *Figure 3A*) is the second-most frequently altered UBIAD1 residue

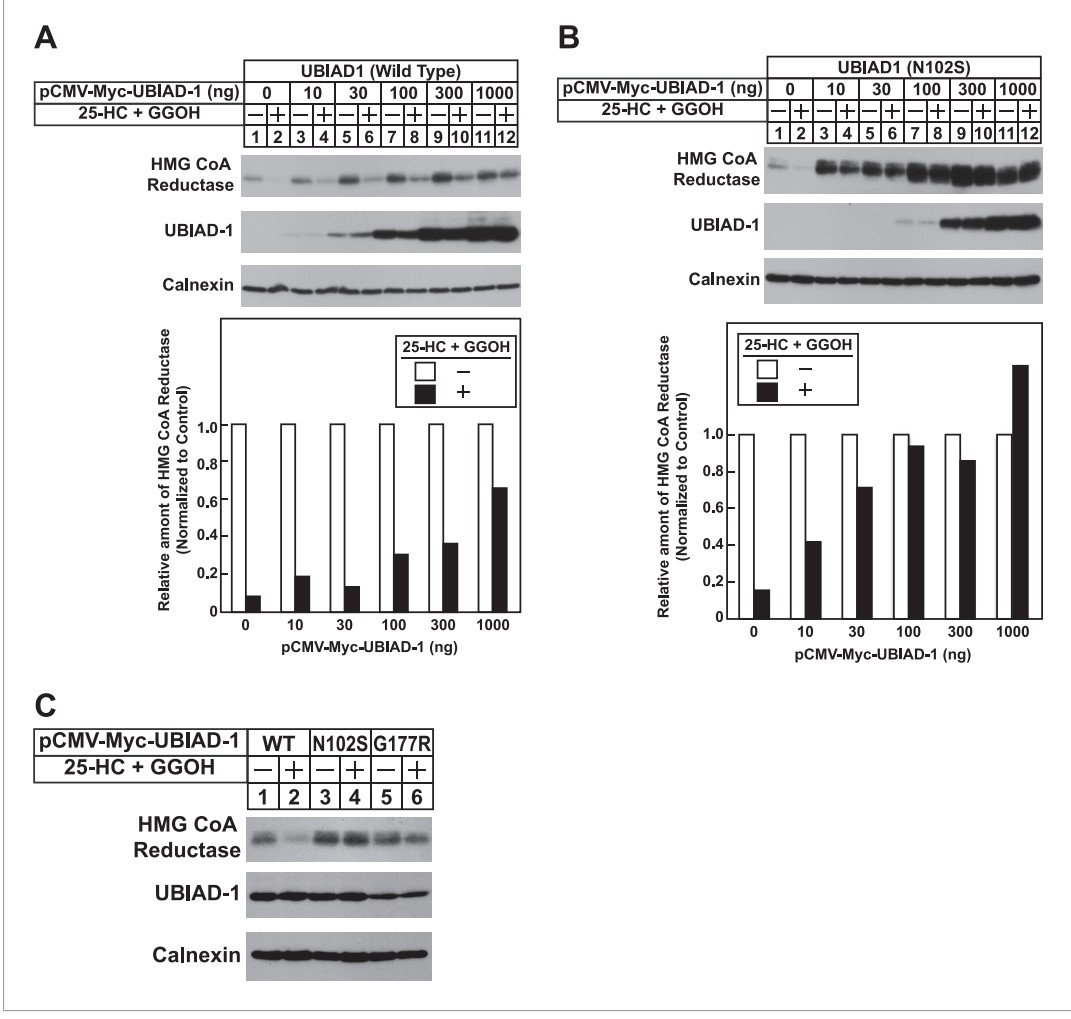

**Figure 7**. The Schnyder corneal dystrophy (SCD)-associated N102S and G177R mutants of UBIAD1 block sterol-accelerated ERAD of HMG CoA reductase. SV-589 cells were set up for experiments on day 0 at $4 \times 10^5$ cells per 60-mm dish in medium A containing 10% FCS. On day 1, cells were transfected with 3 µg/dish of pCMV-HMG-Red(TM1-8)-T7 in the absence or presence of the indicated concentration of pCMV-Myc-UBIAD1 (WT or N102S) (**A** and **B**) or 3 µg/dish of pCMV-HMG-Red(TM1-8)-T7 together with 30 ng of pCMV-Myc-UBIAD1 (WT, N102S, or G177R) (**C**) as described in 'Materials and methods'. 4 hr after transfection, cells received a direct addition of medium A containing 10% NC-LPPS, 10 µM compactin, and 50 µM mevalonate (final concentrations). Following incubation for 16 hr at 37°C, cells were treated with identical medium in the absence or presence of 1 µg/ml 25-HC plus 20 µM geranylgeraniol (GGOH) as indicated. After 4 hr at 37°C, cells were harvested and subjected to subcellular fractionation. Aliquots of resulting membrane fractions were then subjected to SDS-PAGE and immunoblot analysis was carried out with anti-T7 IgG (against reductase), IgG-9E10 (against UBIAD1 and Insgi-1), and anti-calnexin IgG. Proteins corresponding to reductase in (**A** and **B**) were quantified using ImageJ software. The intensities of these signals in the absence of 25-HC plus geranylgeraniol were arbitrarily set as 1.

The following figure supplement is available for figure 7:

**Figure supplement 1**. Schnyder corneal dystrophy (SCD)-associated N102S mutant of UBIAD1 blocks sterol-accelerated ERAD of full-length HMG CoA reductase.

in SCD families (*Nickerson et al., 2013*). The results of *Figure 7C* show that UBAID1 harboring the SCD-associated G177R mutation blocked reductase ERAD to a degree similar to that observed with UBIAD1 (N102S) (top panel, compare lanes 1 and 2 with lanes 3–6).

We next compared the geranylgeraniol-mediated displacement of wild type and N102S UBIAD1 from reductase using co-immunoprecipitation. UBIAD1⁻ cells stably transfected with empty mock

vector, pCMV-Myc-UBIAD1, or pCMV-Myc-UBIAD1 (N102S) were briefly incubated with 25-HC in the absence or presence of geranylgeraniol. The cells were then harvested, lysed, and subjected to anti-reductase immunoprecipitation. The results show that wild type UBIAD1 co-precipitated with reductase in the 25-HC-treated cells (*Figure 8A*, second panel, lane e). This co-precipitation was inhibited by geranylgeraniol (lanes g and h). UBIAD1 (N102S) similarly co-precipitated with reductase (lane i); however, the mutant UBAID1 resisted geranylgeraniol-mediated displacement and remained associated with reductase (lanes j–l). UBIAD1 (N102S) exhibited a similar resistance to geranylgeraniol-mediated displacement from reductase in two independent experiments shown in *Figure 8—figure supplement 1*.

In the experiment of *Figure 8B*, we used immunofluorescence to examine the subcellular localization of wild type and SCD-associated mutants of UBIAD1 in SV-589 cells grown in FCS-containing medium. The results show that wild type UBIAD1 localized to Golgi membranes that also stained with antibodies against GM130 (*Figure 8B*, panels 1–4). UBIAD1 (N102S) primarily exhibited a diffuse, reticular localization corresponding to ER membranes and little, if any, of the protein appeared in the Golgi (*Figure 8B*, panels 5–8). The repeat experiment shown in *Figure 8—figure supplement 2* confirms that wild type UBIAD1 localized to the Golgi, whereas UBIAD1 (N102) primarily localized to ER membranes. Similar to the N102 mutant, UBIAD1 (G177R) was also primarily concentrated in membranes of the ER, but not the Golgi (*Figure 8B*, panels 9–12).

## Discussion

Results of the current experiments form the basis for the model shown in *Figure 9* that depicts the role of UBIAD1 in sterol-accelerated ERAD of reductase. As previously proposed (*Morris et al., 2014*), the reaction is initiated by accumulation of sterols, which triggers binding of Insigs to reductase and results in its gp78/Trc8-mediated ubiquitination (*Song et al., 2005*; *Jo et al., 2011*). We find in the current study that the oxysterol 25-HC and the 1,1-bisphosphonate ester Apomine, which mimics 25-HC in accelerating reductase ERAD (*Roitelman et al., 2004*; *Sever et al., 2004*; *Nguyen et al., 2009*), trigger binding of reductase to UBIAD1 (*Figure 3B,C*), a prenyltransferase that mediates synthesis of the vitamin $K_2$ derivative MK-4 (see *Figure 1*). The sterol-induced binding of UBIAD1 to reductase appears to follow the action of Insigs as indicated by inhibition of the association by RNAi-mediated Insig knockdown (*Figure 4A*). In contrast to the action of sterols, the nonsterol isoprenoid geranylgeraniol, which augments sterol-accelerated reductase ERAD (*Sever et al., 2003a*), inhibits formation of the UBIAD1-reductase complex (*Figure 5A* and *Figure 5—figure supplement 1B*). Geranylgeraniol also regulates the subcellular localization of UBIAD1. The prenyltransferase primarily localizes to Golgi membranes when cells are cultured in sterol-replete FCS-containing medium (*Figure 5C*). The Golgi localization of UBIAD1 is disrupted when cells are switched to sterol-depleting medium containing LPPS and the reductase inhibitor compactin, which depletes cellular stores of nonsterol isoprenoids (*Brown and Goldstein, 1980*). Remarkably, Golgi localization of UBAID1 is restored by the addition of geranylgeraniol, but not of 25-HC or farnesol, to cells deprived of sterol and nonsterol isoprenoids (*Figure 5C* and *Figure 5—figure supplements 2, 3*). We conclude from these observations that geranylgeraniol-mediated disruption of the UBIAD1-reductase complex allows for the translocation of UBIAD1 from ER membranes to the Golgi and for membrane extraction, cytosolic dislocation and proteasomal degradation of reductase (*Figure 9*).

The model shown in *Figure 9* predicts that UBIAD1 inhibits sterol-accelerated ERAD of reductase and that geranylgeraniol relieves this inhibition by triggering displacement of UBIAD1 from the protein. In support of this notion, we observed that RNAi-mediated knockdown and CRISPR/Cas9-mediated knockout of UBIAD1 eliminates the requirement of geranylgeraniol for maximal degradation of reductase (*Figure 6*). The continuance of accelerated reductase degradation in UBIAD1-deficient cells is consistent with the proposal that UBIAD1 mediates a step in the reaction downstream the action of Insigs. In future studies, efforts will be undertaken to delineate mechanisms through which UBIAD1 retards reductase ERAD. Geranylgeraniol is required for extraction of ubiquitinated reductase across the ER membrane through a reaction that is mediated by the AAA-ATPase VCP/p97 (*Morris et al., 2014*). Thus, one likely mechanism for the inhibitory effects of UBIAD1 on reductase ERAD may involve inhibition of the association of ubiquitinated reductase with VCP/p97 or one of its ubiquitin-binding cofactors, thereby preventing its subsequent membrane extraction.

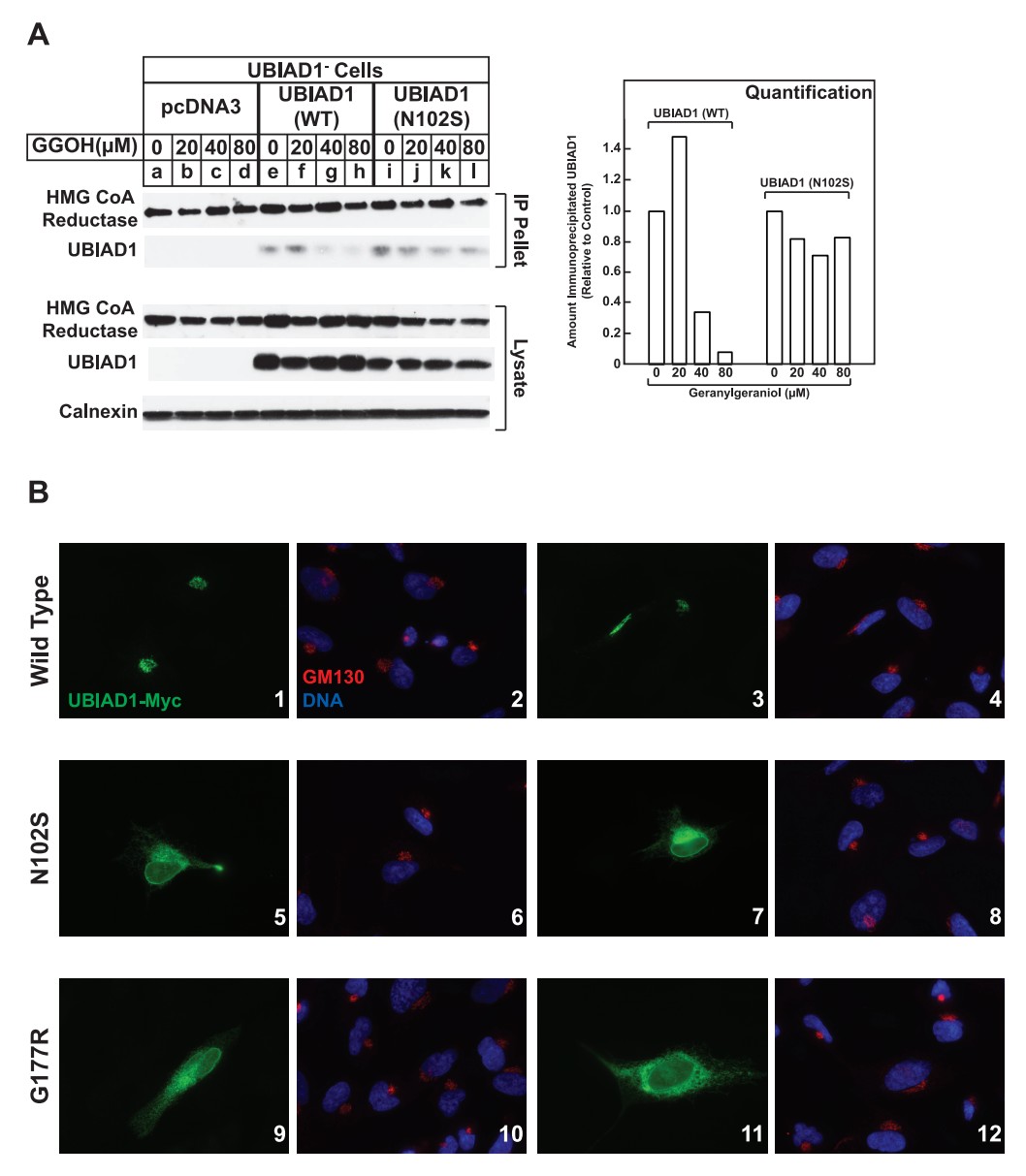

**Figure 8**. SCD-associated UBIAD1 mutant resists geranylgeraniol-mediated displacement from HMG CoA reductase and remains sequestered in ER membranes. (**A**) UBIAD1$^-$/pCDNA3.1, UBIAD1$^-$/pMyc-UBIAD1 (WT), and UBIAD1$^-$/pMyc-UBIAD1 (N102S) cells were set up for experiments on day 0 at a density of 4 × 10$^5$ cells per 60-mm dish in medium A containing 10% FCS. On day 3, cells were depleted of sterols as described in the legend to *Figure 4*. After 16 hr at 37°C, cells received the identical medium containing 1 μg/ml 25-HC in the absence or presence of the indicated concentration of geranylgeraniol. After 45 min at 37°C, cells were harvested, lysed, and immunoprecipitated with polyclonal anti-reductase antibodies. Aliquots of the precipitated material and the lysates were subjected to SDS-PAGE and immunoblot analysis was carried out with IgG-A9 (against reductase), IgG-H8 (against UBIAD1), and anti-calnexin IgG. Proteins corresponding to immunoprecipitated UBIAD1 were quantified using ImageJ software. The intensities of these signals in the absence of geranylgeraniol were arbitrarily set as 1. (**B**) SV-589 cells were set up on day 0 at 3 × 10$^4$ cells/well of a twelve-well plate with a glass coverslip in medium A containing 10% FCS. On day 1, the cells were transfected using FuGENE 6 with 50 ng of wild type (WT), N102S, or G177R versions of pCMV-Myc-UBIAD1; the total amount of DNA/well was adjusted to 500 ng by the addition of pcDNA3.1 vector. 4 hr after transfection, cells received a direct addition of medium A containing 10% FCS

*Figure 8. Continued*

(final concentration). After 16 hr at 37°C, cells were fixed and analyzed by microscopy as described in 'Materials and methods'.
The following figure supplements are available for figure 8:

**Figure supplement 1**. SCD-associated UBIAD1 (N102S) resists geranylgeraniol-mediated displacement from HMG CoA reductase in two independent experiments (**A** and **B**).
**Figure supplement 2**. Subcellular localization of wild type and SCD-associated N102S UBIAD1 in transfected SV-589 cells.

In addition to revealing the molecular mechanism through which geranylgeraniol modulates reductase ERAD, the current results help to explain how cells maintain synthesis of nonsterol isoprenoids, while avoiding overproduction of cholesterol. In 1979, Faust, Brown, and Goldstein found that in the absence of LDL, cells accumulate reductase to produce mevalonate for conversion

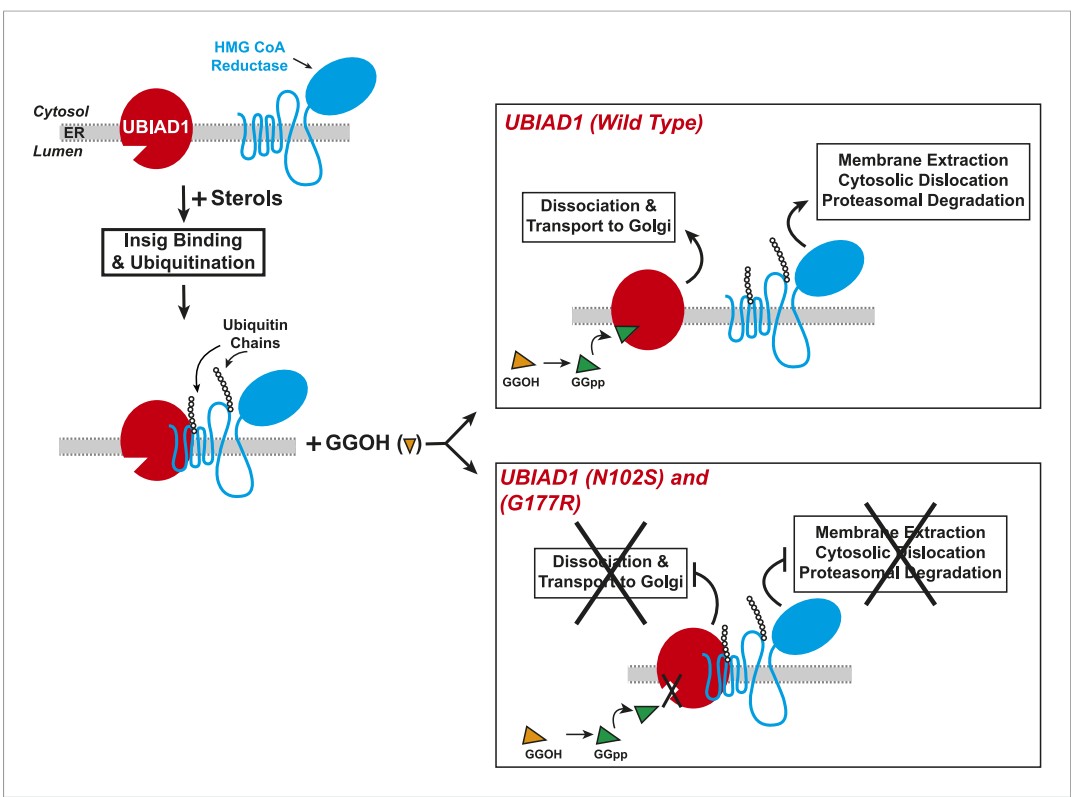

**Figure 9**. Proposed model for role of UBIAD1 in sterol-accelerated degradation of HMG CoA reductase. In sterol-deprived cells, both reductase and UBIAD1 localize to membranes of the ER. The intracellular accumulation of sterols in ER membranes triggers binding of reductase to Insigs, resulting in its ubiquitination by Insig-associated ubiquitin ligases gp78 and Trc8 and association with UBIAD1. Geranylgeraniol becomes phosphorylated to produce geranylgeranyl pyrophosphate, which enhances reductase degradation by binding to UBIAD1, causing its displacement from reductase-Insig. This displacement allows for transport of UBIAD1 to the Golgi and membrane extraction, cytosolic dislocation, and proteasomal degradation of reductase. We postulate that the SCD-associated N102S or G177R mutations in UBIAD1 abrogate binding of geranylgeranyl pyrophosphate. As a result, UBIAD1 (N102S) and (G177R) do not translocate to the Golgi and remain associated with reductase in the ER, thereby blocking its membrane extraction, cytosolic dislocation, and proteasomal degradation.

into primarily cholesterol and other sterols (*Faust et al., 1979*). The addition of LDL to cells partially suppressed reductase, limiting production of mevalonate. As a result, incorporation of mevalonate into cholesterol was reduced, whereas incorporation of the molecule into nonsterol isoprenoids such as CoQ10 was enhanced. We postulate that LDL failed to completely suppress reductase due to its binding to UBIAD1 and resultant inhibition of sterol-accelerated degradation. This residual reductase produces a small amount of mevalonate that is preferentially diverted into nonsterol isoprenoids. Once appropriate levels of geranylgeranyl pyrophosphate accumulate in cells, the isoprenoid binds to UBIAD1, causing its release from reductase, which subsequently becomes degraded. Released UBIAD1 then translocates to the Golgi where it synthesizes PPHB (*Mugoni et al., 2013*) and perhaps MK-4. Thus, substrate-regulated transport of UBIAD1 ensures that products of the enzyme are only produced when cellular demands for other nonsterol isoprenoids as well as sterols have been met. Studies are currently underway to appraise this notion and to determine whether Golgi-localized UBIAD1 is enzymatically active to synthesize MK-4 and PPHB, whereas the ER-localized enzyme is not. Moreover, the role of reductase in ER to Golgi trafficking of UBAD1 will be investigated.

The significance of geranylgeraniol-mediated displacement of UBIAD1 from reductase is revealed through the study of SCD-associated mutants of the prenyltransferase. Consistent with the autosomal dominant phenotype of SCD, disease-associated N102S and G177R mutants of UBIAD1 inhibit sterol-accelerated ERAD of reductase (*Figure 7B,C*, and *Figure 7—figure supplement 1*). The molecular basis for this inhibition is suggested by *Figure 8A* and *Figure 8—figure supplement 1*, which show that UBIAD1 (N102S) resists geranylgeraniol-mediated displacement from reductase. Subcellular localization studies show that in FCS-cultured cells, wild type UBIAD1 localizes to the Golgi, whereas the N102S and G177R mutants of the enzyme appear defective in Golgi trafficking and remain sequestered within the ER (*Figure 8B*). Mutation of the asparagine residue in bacterial UbiA prenyltransferases that corresponds to N102 markedly diminishes substrate binding and enzymatic activity (*Cheng and Li, 2014*; *Huang et al., 2014*). Thus, we postulate that binding of UBIAD1 to geranylgeranyl pyrophosphate produced by phosphorylation of geranylgeraniol triggers displacement of wild type UBIAD1 from reductase, allowing the protein to translocate to the Golgi. In contrast, the N102S and G177R mutants of UBIAD1 exhibit reduced affinity for geranylgeranyl pyrophosphate and thus, remain associated with reductase in the ER, thereby inhibiting its sterol-accelerated ERAD (*Figure 9*). This inhibition may contribute to the accumulation of cholesterol that is observed in corneas of SCD patients. It will be important in future studies to develop a geranylgeranyl pyrophosphate-binding assay for mammalian UBIAD1 and to determine the subcellular localization and effect of the other 18 SCD-associated UBIAD1 mutants on reductase ERAD.

In conclusion, the current study has identified UBIAD1 as the elusive molecular target of geranylgeraniol in reductase ERAD. They have also revealed a potential mechanism whereby mutations in UBIAD1 cause the accumulation of cholesterol in corneas of SCD patients. Finally, results of this study indicate the existence of a novel link between the synthesis of MK-4 and cholesterol. Further investigation of this link is merited owing to the potential for development of novel therapies that accelerate reductase ERAD to lower plasma LDL-cholesterol and retard corneal accumulation of cholesterol that characterizes SCD.

## Materials and methods

### Materials

We obtained MG-132 from Boston Biochem (Cambridge, MA); horseradish peroxidase-conjugated donkey anti-mouse and anti-rabbit IgGs (affinity-purified) were from Jackson ImmunoResearch Laboratories (West Grove, PA); digitonin was from Calbiochem (San Diego, CA); geranylgeraniol and farnesol from Sigma–Aldrich (St. Louis, MO) and Santa Cruz Biotechnology (Dallas, TX); biotin was obtained from Sigma–Aldrich (St. Louis, MO); and 25-hydroxycholesterol from Steraloids (Newport, RI). Apomine was synthesized by the Core Medicinal Chemistry laboratory at the University of Texas Southwestern Medical Center. Other reagents, including new born calf lipoprotein-poor serum (NC-LPPS, d > 1.215 g/ml), sodium compactin, sodium mevalonate, and stock solutions of digitonin were prepared or obtained from previously described sources (*Goldstein et al., 1983*; *DeBose-Boyd et al., 1999*; *Elsabrouty et al., 2013*).

## Expression plasmids

The previously described expression plasmids pCMV-HMG-Red(TM1-8)-T7 and pCMV-HMG-Red-T7 encode the membrane domain (amino acids 1–346) and full-length (amino acids 1–887) forms, respectively, of hamster reductase followed by three tandem copies of the T7 epitope tag under transcriptional control of the cytomegalovirus (CMV) promoter (Sever et al., 2003a, 2003b). The expression plasmid pCMV-HSV-HMG-Red(TM1-8)-BirA* encodes the membrane domain of hamster reductase with two copies of an N-terminal epitope tag derived from Herpes Simplex Virus (HSV) glycoprotein D fused to the humanized R118G mutant of *Escherichia coli* BirA (obtained from Addgene, Cambridge, MA and designated BirA*) that exhibits promiscuous biotin ligase activity (Roux et al., 2012). The cDNA encoding human UBIAD1 was purchased from Open Biosystems (Lafayette, CO) and cloned into the pcDNA3.1(+) vector using standard PCR methods. The expression plasmid pCMV-Myc-UBIAD1 was generated by fusing one copy of the Myc epitope tag to the N-terminus of UBIAD1. The plasmids pCMV-Myc-UBIAD1 (N102S) and (G177R) encode Myc-tagged human UBIAD1 harboring the SCD-associated asparagine-102 to serine (N102S) and glycine-177 to arginine (G177R) mutations, respectively, and were generated using the Quikchange Site-Directed Mutagenesis Kit (Agilent Technologies, Santa Clara, CA) and pCMV-Myc-UBIAD1 as a template. CRISPR plasmids hCas9 and gRNA Cloning Vectors were obtained from Addgene. Guide RNA constructs were designed using option B described by the Church laboratory (Mali et al., 2013). (See http://www.addgene.org/static/cms/files/hCRISPR_gRNA_Synthesis.pdf) Guide RNA sequences unique to human UBIAD1 were selected from a published list (Mali et al., 2013). (See http://arep.med.harvard.edu/human_crispr).

## Cell culture

SV-589 cells are a line of immortalized human fibroblasts expressing the SV40 large T-antigen (Yamamoto et al., 1984). Monolayers of SV-589 cells were maintained in medium A (DMEM containing 1000 mg glucose/l, 100 U/ml penicillin, and 100 mg/ml streptomycin sulfate) supplemented with 10% (vol/vol) fetal calf serum (FCS) at 37°C, 5% $CO_2$. Human embryonic kidney (HEK)-293S/pHMG-Red(TM1-8)-BirA* cells were generated as follows: on day 0, HEK-293S cells were set up at a density of $7 \times 10^5$ cells per 100-mm dish in medium A supplemented with 10% FCS. On day 1, cells were transfected with 6 µg/dish of pCMV-HSV-HMG-Red(TM1-8)-BirA* using FuGENE6 transfection reagent (Promega, Madison, WI) as previously described (Sever et al., 2003b; Jo et al., 2011). Following incubation for 16 hr at 37°C, cells were switched to medium A supplemented with 10% FCS and 700 µg/ml G418. Fresh medium was added every 2–3 days until colonies formed after 2 weeks. Individual colonies were isolated using cloning cylinders, and expression of HSV-HMG-Red (TM1-8)-BirA* was determined by immunoblot analysis. Cells from single colonies expressing high levels of HSV-HMG-Red(TM1-8)-BirA* were selected and monolayers were maintained in medium B (medium A supplemented with 10% FCS and 700 µg/ml G418) at 37°C, 5% $CO_2$.

UBIAD1-deficient cells (designated UBIAD1⁻) were generated as follows: on day 0, SV589 cells were set up at a density of $7 \times 10^5$ cells per 100-mm dish in medium A supplemented with 10% FCS. On day 1, cells were transfected with 5 µg/dish each of hCas9, hUBIAD1-gRNA12 and hUBIAD1-gRNA19 using FuGENE6 transfection reagent as described above. On day 2 and 3 the transfection above was repeated. On day 4 cell clones were isolated using serial dilution in 96-well plates. Clones were screened for the absence of UBIAD1 by immunoblot analysis using mouse monoclonal IgG-H8 and rabbit polyclonal antibodies against human UBIAD1 (Santa Cruz Biotechnology, Dallas, TX). A homozygous 113 bp deletion/frameshift mutation (starting at codon 60) of UBIAD1 was identified by PCR and sequencing of the PCR products by standard techniques. UBIAD1⁻/pcDNA3.1, UBIAD1⁻/pMyc-UBIAD1, and UBIAD1⁻/pMyc-UBIAD1 (N102S) are UBIAD1⁻ cells stably transfected with pcDNA3.1, pCMV-Myc-UBIAD1, and pCMV-Myc-UBIAD1 (N102S), respectively. These cells were generated as follows: on day 0, UBIAD1⁻cells were set up at a density of $7 \times 10^5$ cells per 100-mm dish in medium A supplemented with 10% FCS. On day 1, cells were transfected with 6 µg/dish of pcDNA3.1, pCMV-Myc-UBIAD1, or pCMV-Myc-UBIAD1 (N102S) using FuGENE6 transfection reagent as described above. Following incubation for 16 hr at 37°C, cells were switched to medium A supplemented with 10% FCS and 700 µg/ml G418. Fresh medium was added every 2–3 days until colonies formed after 2 weeks. Individual colonies were isolated using cloning cylinders, and expression of Myc-UBIAD1 was determined by immunoblot analysis. Cells from single colonies of cells expressing moderate levels of

Myc-UBIAD1 were selected and monolayers were maintained in medium B (medium A supplemented with 10% FCS and 700 µg/ml G418) at 37°C, 5% $CO_2$.

## Isolation of HMG-Red(TM1-8)-BirA*-associated proteins

HEK-293S/pHMG-Red(TM1-8)-BirA* cells were set up on day 0 at a density of $3 \times 10^5$ cells per 100-mm dish in medium B. On day 3, cells were depleted of sterols through incubation in medium A supplemented with 10% (vol/vol) NC-LPPS, 10 µM compactin, and 50 µM mevalonate. After 16 hr at 37°C, the cells were treated for an additional 6 hr in sterol-depleting medium containing 50 µM biotin and 1 µg/ml 25-HC. Cells were subsequently harvested, washed with phosphate-buffered saline (PBS), and resuspended in 0.3 ml/plate of buffer containing 10 mM HEPES, pH 7.4, 10 mM KCl, 1.5 mM $MgCl_2$, 5 mM sodium EDTA, 5 mM sodium EGTA, and 250 mM sucrose supplemented with a protease inhibitor cocktail consisting of 5 mM dithiothreitol, 0.1 mM leupeptin, 1 mM phenylmethylsulfonyl fluoride, 0.5 mM Pefabloc, 5 µg/ml pepstatin A, 25 µg/ml ALLN, and 10 µg/ml aprotinin. The cell suspension was passed through a 22.5 gauge needle 25 times and centrifuged at 1000×g for 10 min at 4°C. The resultant post-nuclear supernatant was subjected to an additional round of centrifugation at 100,000×g for 30 min at 4°C. The membrane pellet of this spin was resuspended in 0.3 ml/10 plates of solubilization buffer containing 50 mM Tris-HCl, pH 7.4, 150 mM NaCl, 5 mM EDTA, 5 mM EGTA, 8 M urea, and 2% SDS and subjected to dounce homogenization. The solubilized membrane pellet was then precleared with 1 ml/10 plate of agarose beads (Sigma–Aldrich, St. Louis, MO) by rotation for 6–16 hr at 4°C. After centrifugation at 300×g for 5 min at 4°C, the supernatants were mixed with 0.8 ml of streptavidin-coupled agarose beads (Sigma–Aldrich, St. Louis, MO) and rotated overnight at 4°C. The beads were collected by centrifugation at 2500×g for 1.5 min at 4°C, and washed three times with three volumes of solubilization buffer diluted 10-fold in Tris-HCl, pH 7.4, 150 mM NaCl, 5 mM EDTA, and 5 mM EGTA. The streptavidin-coupled beads were then eluted three times through incubation for 10 min at 98°C in 1 volume of buffer containing 50 mM Tris-HCl, pH 7.4, 150 mM NaCl, 5 mM EGTA, 5 mM EDTA, 5 mM biotin, 10% (vol/vol) dimethylsulfoxide, 2% SDS, and 0.4 M urea. The elutions were pooled and precipitated with 4 vol of ice-cold acetone at −20°C for 1 hr. Precipitated material was collected by centrifugation at 16,000 at 4°C and dissolved in a mixture consisting of 10 µl of buffer containing 10 mM Tris-HCl, pH 6.8, 100 mM NaCl, 1% SDS, 1 mM EDTA, and 1 mM EGTA; 20 µl of buffer containing 62.5 mM Tris-HCl, pH 6.8, 15% SDS, 8 M urea, 10% glycerol, and 100 mM dithiothreitol; 10 µl 4× SDS-PAGE loading buffer. The samples were incubated for 20 min at 37°C prior to SDS-PAGE. The gels were subjected to staining with colloidal coomassie as described (*Dyballa and Metzger, 2009*) and three segments of the gel that contained visible bands (see *Figure 3—figure supplement 1A*) were excised, and the identities of the proteins were determined by tandem mass spectrometry in the Protein Chemistry Core Facility at the University of Texas Southwestern Medical Center.

## RNA interference

RNA interference (RNAi) was performed as previously described with minor modifications (*Jo et al., 2011*). Duplexes of siRNAs were designed and synthesized by Dharmacon/Thermo Fisher Scientific. The siRNA duplexes against GFP, Insig-1, and Insig-2 have been previously described (*Elsabrouty et al., 2013*); the sequence for siRNA duplexes targeting human UBIAD1 and reductase were 5′-UUAA CAUCCUGUCGGGAGAUU-3′ and 5′-CCACAGAGGCUAUGAUUGAUU-3′, respectively. SV-589 cells were set up for experiments on day 0 as described in the figure legends. On day 3, the cells were incubated with 600 pmol of siRNA duplexes mixed with Lipofectamine RNAiMAX reagent (Invitrogen, Grand Island, NY) diluted in Opti-MEM I reduced serum medium (Life Technologies, Grand Island, NY) according to manufacturer's procedure. Following incubation for 6 hr at 37°C, the cells received a direct addition of medium A containing 10% FCS (final concentration). On day 4, the cells were switched to medium A containing 10% NC-LPPS, 10 µM compactin, and 50 µM mevalonate and incubated for 16 hr at 37°C. The cells were subsequently treated and analyzed as described in figure legends.

## Transfection, subcellular fractionation, immunoprecipitation, and immunoblot analysis

Transfection of SV-589 cells with FuGENE6 transfection reagent was carried out as previously described (*Sever et al., 2003b*; *Jo et al., 2011*). Conditions of subsequent incubations are described in the figure legends. Following incubations, triplicate dishes of cells for each variable were harvested and pooled for analysis. For immunoprecipitations with polyclonal anti-reductase or anti-UBIAD1

antibodies, the cells were resuspended in PBS containing 1% digitonin, 5 mM EDTA, 5 mM EGTA, and the protease inhibitor cocktail. Following passage through a 22.5 gauge needle 15 times and rotation for 30 min at 4°C, the samples were clarified by centrifugation at 20,000×$g$ for 10 min at 4°C. The detergent-solubilized material was then subjected to immunoprecipitation as described previously (*Elsabrouty et al., 2013*). Subcellular fractionation of cells by differential centrifugation was performed as previously described (*Jo et al., 2013*). Aliquots of detergent lysates (0.01 dish of cells) and pellet fractions (0.125 dish of cells) from immunoprecipitations and membrane fractions from subcellular fractionations were subjected to SDS-PAGE and immunoblot analysis. Primary antibodies used for immunoblot analysis included: mouse monoclonal anti-T7 Tag IgG and anti-HSV Tag (Novagen, Darmstadt, Germany); IgG-A9, a mouse monoclonal antibody against the catalytic domain of reductase (*Liscum et al., 1983*); rabbit polyclonal anti-UBIAD1 IgG, mouse monoclonals IgG-H8 against UBIAD1, IgG-F2 against human peroxiredoxin-4, IgG-F3 against human progesterone receptor membrane component 2 (PGRMC2), IgG-C6 against human ERGIC-53, IgG-ANNEX 5E4/1 against human annexin I, and IgG-Y20 against human lamina-associated peptide-2 (Santa Cruz Biotechnology, Dallas, TX); rabbit polyclonal anti-calnexin IgG (Novus Biologicals, Littleton, CO); and IgG-17H1, a mouse monoclonal antibody against human Insig-1.

## Immunofluorescence

SV-589 cells were set up for experiments on day 0 as described in the figure legends. Following incubations described in the figure legends, cells were washed with PBS and subsequently fixed and permeabilized for 15 min in methanol at −20°C. Upon blocking with 1 mg/ml BSA in PBS, coverslips were incubated for 30 min at 37°C with primary antibodies (IgG-H8 against UBIAD1, rabbit polyclonal anti-GM130 IgG [*Diao et al., 2003*], and IgG-9E10, a mouse monoclonal antibody against c-Myc purified from the culture medium of hybridoma clone 9E10 [American Type Culture Collection, Manassas, VA]) diluted in PBS containing 1 mg/ml BSA. Bound antibodies were visualized with goat anti-mouse IgG conjugated to Alexa Fluor 488 and goat anti-rabbit Alexa Fluor 594 (Life Technologies, Grand Island, NY) as described in the figure legends. In addition, coverslips were stained for 5 min with 1 µg/ml Hoechst 33,342 (Life Technologies) to visualize nuclei. The coverslips were then mounted in Mowiol 4-88 solution (Calbiochem/EMD Millipore, Billerica, MA) and fluorescence analysis was performed using a Plan-Apochromat 63×/1.4 DIC objective (Zeiss, Peabody, MA), an Axiovert 200M microscope (Zeiss), an Orca 285 camera (Hamamatsu,Houston, TX), and the software Openlab 4.0.2 (Improvision, Coventry, England).

## Acknowledgements

We thank Drs Michael S Brown and Joseph L Goldstein for their continued encouragement and advice; Dr David W Russell for critical evaluation of the manuscript; we also thank Lisa Beatty, Muleya Kapaale, Hue Dao, Loren Valsin, and Ijeoma Dukes for help with tissue culture.

## Additional information

### Funding

| Funder | Grant reference | Author |
| --- | --- | --- |
| Howard Hughes Medical Institute (HHMI) | Early Career Scientist | Russell A DeBose-Boyd |
| National Institutes of Health (NIH) | HL20948-37 | Russell A DeBose-Boyd |
| National Institutes of Health (NIH) | GM112409-01 | Russell A DeBose-Boyd |

The funders had no role in study design, data collection and interpretation, or the decision to submit the work for publication.

### Author contributions

MMS, RE, Conception and design, Acquisition of data, Analysis and interpretation of data, Drafting or revising the article; JS, Conception and design, Acquisition of data, Analysis and interpretation of data; YJ, Conception and design, Drafting or revising the article, Contributed unpublished essential

data or reagents; RADB, Conception and design, Analysis and interpretation of data, Drafting or revising the article

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
