## [Decision Letter]

Thank you for sending your work entitled “UBIAD1 is the target of geranylgeraniol in degradation of HMG CoA reductase: Implications for Schnyder Corneal Dystrophy” for consideration at *eLife*. Your article has been favorably evaluated by Randy Schekman (Senior editor) and 3 reviewers, one of whom is a member of our Board of Reviewing Editors.

The Reviewing editor and the other reviewers discussed their comments before we reached this decision, and the Reviewing editor has assembled the following comments to help you prepare a revised submission.

This study presents a series of experiments that identifies UBIAD1 as an HMG CoA reductase interactor that regulates its degradation in a geranylgeraniol (GGOH)-dependent manner. Dominant mutations in UBIAD1 that cause Schnyder corneal dystrophy (SCD) disrupt this regulation. The work resolves a long-standing question of how GGOH enhances the sterol-dependent degradation of mammalian HMGR, which has been a consistent yet mysterious feature of HMGR degradation as studied by this group for a number of years. The results make sense, and lead to a number of hypotheses and perhaps distinct therapeutic approaches for Schnyder Corneal Dystrophy. The topic and findings should be of interest to the readers of *eLife* and open up several future areas for study. Nevertheless, some aspects of the study require clarification and a modest amount of new experimentation to fully support the authors' conclusions. The following three points were deemed essential to the central conclusions, while the 'minor comments' below are more peripheral.

1) Figure 6 is the central result, and it is worth making it completely convincing by presenting a time course of reductase degradation in wild type versus UBIAD knockout cells. Ideally, this would be done by radiolabeled pulse-chase, but even a western blot at different times after 25-HC or Apomine treatment would provide a clear picture of what is happening. This is important because the steady state levels of reductase are different in wild type and knockout cells, complicating the comparison. Thus, a carefully controlled and quantified time course would help to directly show different rates of degradation.

2) In Figure 7, the authors analyze the levels of transfected HMG membrane domain (residues 1-346) and not the endogenous protein. This is important since the effect of WT and mutant UBIAD1 on this truncated form (Figure 7) and on endogenous reductase (Figure 8, Lysate) appears quite different. While the steady state levels of the truncated protein significantly increase upon UBIAD1 transfection, the endogenous protein does not seems to change much. Can the authors please explain the rationale for using the truncated membrane domain, or if possible, present data with the full-length protein?

3) Due perhaps to the relatively modest effects of UBIAD on reductase, there appears to be a fair amount of sample-to-sample variability (as noted in several of the 'Minor comments' below). While consistent trends from multiple different approaches make an overall convincing case, certain individual experiments were less convincing to some reviewers. In particular, the key result of Figure 8 is not totally convincing and quantification of independent experiments would help to validate the conclusion that N102S is GGOH-insensitive.

Minor comments:

1) The reductase-BirA* strategy for identifying interacting partners should be documented. The approach is clearly useful for finding membrane protein interactors, and it is likely that readers will want to apply this to other systems if it is robust. Unless there is a compelling reason otherwise, I recommend including a figure and appropriate supplemental table documenting the interactors identified and where among these UBIAD1 ranked. It would be worth noting if known interactors (e.g., Insigs and ubiquitin ligases) were observed as expected.

2) In Figure 3, reductase levels increase substantially during the time course, even with 25-HC. It eventually seems to be degraded in a 25-HC-dependent manner, although the level at 240 min seems higher (in the IP sample) or slightly lower (in the lysate) relative to starting amount. Yet, in later experiments (Figure 6), 4 hours with 25-HC results in what appears to be ∼90% degradation. Please clarify this apparent discrepancy.

3) In many of the figures, relative reductase levels in the starting lysate and after the IP are different. As one example, lanes 1-3 in Figure 5 show substantial reduction in reductase with 25-HC in lysate, but an apparent increase in the IP samples. Is this simply experimental variability or is there some other reason (e.g., antibody accessibility, differences in ubiquitination levels, etc.)? Please comment on this in the text or figure legend so a reader is not confused.

4) The authors discuss a role for UBIAD1 after Insig binding and ubiquitination. However, UBIAD1 knockdown seems to substantially reduce reudctase-Insig interaction (Figure 4). Please clarify why they propose UBIAD1 acting late, and if the reason(s) are not convincing, some caveat is warranted.

---

## [Author Response]

*1)*
Figure 6
*is the central result, and it is worth making it completely convincing by presenting a time course of reductase degradation in wild type versus UBIAD knockout cells. Ideally, this would be done by radiolabeled pulse-chase, but even a western blot at different times after 25-HC or Apomine treatment would provide a clear picture of what is happening. This is important because the steady state levels of reductase are different in wild type and knockout cells, complicating the comparison. Thus, a carefully controlled and quantified time course would help to directly show different rates of degradation*.

We thank the reviewers for this suggestion. A new experiment (Figure 6) is now included in the revised manuscript. This experiment shows a time course of the effects of geranylgeraniol on accelerated degradation of reductase in wild type and UBIAD1 knockout cells.

*2) In*
Figure 7*, the authors analyze the levels of transfected HMG membrane domain (residues 1-346) and not the endogenous protein. This is important since the effect of WT and mutant UBIAD1 on this truncated form (*Figure 7*) and on endogenous reductase (*Figure 8*, Lysate) appears quite different. While the steady state levels of the truncated protein significantly increase upon UBIAD1 transfection, the endogenous protein does not seems to change much. Can the authors please explain the rationale for using the truncated membrane domain, or if possible, present data with the full-length protein?*

The experiment of Figure 7 was designed to determine the effect of UBIAD1 (N102S) on sterol-accelerated degradation of reductase. We utilized the truncated membrane domain of reductase because this region of the protein is both necessary and sufficient for sterol-accelerated degradation (Skalnik et al., 1988, J. Biol. Chem., and Gil et al., 1985, Cell). We have observed in unpublished studies that the membrane domain of reductase appears to be more susceptible to accelerated degradation than the full-length form, which could explain why UBIAD1 stabilizes the membrane domain of reductase more than the full-length protein. The difference in stability between the membrane domain of reductase and the full-length protein may be attributable to the C-terminal catalytic domain, which is known to form a tetramer that likely has to be unassembled prior to proteasomal degradation. As suggested by the reviewers, we have now included experiments (Figure 7—figure supplement 1) that compare the ability of wild type and mutant (N102S) UBIAD1 to block degradation of full-length reductase. Results similar to those with the truncated membrane domain of reductase were obtained with the full-length enzyme.

*3) Due perhaps to the relatively modest effects of UBIAD on reductase, there appears to be a fair amount of sample-to-sample variability (as noted in several of the 'Minor comments' below). While consistent trends from multiple different approaches make an overall convincing case, certain individual experiments were less convincing to some reviewers. In particular, the key result of*
Figure 8
*is not totally convincing and quantification of independent experiments would help to validate the conclusion that N102S is GGOH-insensitive*.

We agree with the reviewers’ assessment that the effects of geranylgeraniol on displacement of N102S UBIAD1 from reductase are modest. Thus, we have included in the revised manuscript an additional two independent experiments that demonstrate UBIAD1 (N102S) is more resistant to geranylgeraniol-mediated displacement from reductase than its wild type counterpart (see Figure 8—figure supplement 1). We previously showed that reductase remains in the ER when cells are treated with sterol and nonsterol isoprenoids (see Hartman et al., 2010, J. Biol. Chem, 285:19288). The current observations show that: 1) geranylgeraniol triggers translocation of UBIAD1 to the Golgi in sterol and nonsterol isoprenoid-depleted cells; and 2) UBIAD1 (N102S) remains sequestered in ER membranes. Thus, we conclude the most likely mechanism of action for UBIAD1 in reductase degradation involves its geranylgeraniol-mediated displacement from reductase, which allows reductase degradation and translocation of UBIAD1 to the Golgi. UBIAD1 (N102S) exhibits reduced affinity for geranylgeraniol (or geranylgeranyl pyrophosphate) and remains associated with reductase in the ER, thereby blocking its degradation.

*Minor comments*:

*1) The reductase-BirA* strategy for identifying interacting partners should be documented. The approach is clearly useful for finding membrane protein interactors, and it is likely that readers will want to apply this to other systems if it is robust. Unless there is a compelling reason otherwise, I recommend including a figure and appropriate supplemental table documenting the interactors identified and where among these UBIAD1 ranked. It would be worth noting if known interactors (e.g., Insigs and ubiquitin ligases) were observed as expected*.

We thank the reviewers for this suggestion. We have now included in the revised manuscript the documentation of proteins identified as associated proteins of the reductase-BirA* chimera. As shown in Figure 3—figure supplement 1, we excised three segments from a colloidal blue-stained gel of proteins that co-purified with reductase-BirA*. These segments were subjected to trypsin digestion and the proteins were identified by mass spectrometry. We identified 15 proteins from these segments that gave spectral counts greater 4; these proteins are indicated in the figure. In the experiment shown in Figure 3—figure supplement 1, we used co-immunoprecipitation to confirm the association of the identified protein with endogenous reductase. Some of the proteins did not associate with reductase, while others bound in both the absence and presence of 25-hydroxycholesterol (25-HC). UBIAD1 was the only protein that exhibited sterol-regulated association with endogenous reductase. This feature, along with a previous report for an association between reductase and UBIAD1, and accumulation of cholesterol in SCD, prompted us to focus attention on UBIAD1. We did not identify Insigs or the ubiquitin ligases that mediate reductase ubiquitination in the three segments of the gel that were selected for protein identification. The molecular weights of the selected segments did not correspond to those of the ubiquitin ligases.

*2) In*
Figure 3*, reductase levels increase substantially during the time course, even with 25-HC. It eventually seems to be degraded in a 25-HC-dependent manner, although the level at 240 min seems higher (in the IP sample) or slightly lower (in the lysate) relative to starting amount. Yet, in later experiments (*Figure 6*), 4 hours with 25-HC results in what appears to be ∼90% degradation. Please clarify this apparent discrepancy*.

In the revised manuscript, we have replaced the previous time-course experiment with a new one that measures the association of UBIAD1 with reductase. The results show that UBIAD1 appears in the reductase immunoprecipitates after 30 min of treatment and continues to co-precipitate with reductase at later times, even though total reductase is reduced by accelerated degradation.

*3) In many of the figures, relative reductase levels in the starting lysate and after the IP are different. As one example, lanes 1-3 in*
Figure 5
*show substantial reduction in reductase with 25-HC in lysate, but an apparent increase in the IP samples. Is this simply experimental variability or is there some other reason (e.g., antibody accessibility, differences in ubiquitination levels, etc.)? Please comment on this in the text or figure legend so a reader is not confused*.

We believe some of these differences may result from experimental variability or by slight differences in rates of degradation among different experiments. We attempted to eliminate this variability by blocking degradation of reductase using the proteasome inhibitor MG-132; however, as shown in Figure 3—figure supplement 1, MG-132 blocks degradation of reductase, but inhibits its binding to UBIAD1. It should also be noted that we have observed sterol- or Apomine-induced binding of UBIAD1 to reductase in at least 10 independent experiments of the current manuscript. To further confirm that geranylgeraniol inhibits sterol-induced binding of reductase to UBIAD1, we have included a repeat experiment shown in Figure 3—figure supplement 1.

*4) The authors discuss a role for UBIAD1 after Insig binding and ubiquitination. However, UBIAD1 knockdown seems to substantially reduce reudctase-Insig interaction (*Figure 4*). Please clarify why they propose UBIAD1 acting late, and if the reason(s) are not convincing, some caveat is warranted*.

Figure 4 shows that knockdown of UBIAD1 reduces the sterol-induced binding of reductase to Insig. This suggests that UBIAD1 stabilizes the reductase-Insig complex, but is not absolutely required for the interaction. It should be noted that in the absence of UBIAD1, sterols continue to accelerate degradation of reductase. This indicates that UBIAD1 acts downstream Insig in the degradation of reductase and is not absolutely required for the reaction. Moreover, data indicate that geranylgeraniol modulates the subcellular localization of UBIAD1 and considering that geranylgeraniol mediates a post-ubiquitination step in reductase degradation, we conclude the action of UBIAD1 follows that of Insig.